# Preference Banzhaf: A Game-Theoretic Index with Feature-wise Probabilities

## Abstract

Game-theoretic feature attribution methods are popular in XAI because they satisfy several desirable axioms. Approximating a model as a game with input features as players, these methods measure the weighted average contribution of each feature to a model's prediction across different feature subsets. However, these techniques also make strict assumptions that may affect the quality of the explanations. One common assumption is that all features can join or leave a subset with probability of 0.5, i.e., all subsets are equally likely to form. However, in real games, each player can have different preference for joining a coalition, shifting the probability of the subsets and thus the attribution values. Following this notion, we introduce Preference Banzhaf, which calculates Banzhaf-like value with adjusted probabilities using centered linear regression. We theoretically show the convergence of Preference Banzhaf and empirically demonstrate the effect of probability adjustment on explanation quality and sensitivity.

## 1 Introduction

Artificial Intelligence (AI) is becoming a ubiquitous tool in many fields thanks to their capacity to reflect complicated patterns in large datasets. However, this capacity is often accompanied by high model complexity, making it difficult to interpret a model's prediction process. In high-stakes domains like health care or finance [1, 2], interpretability is as important as the accuracy of prediction, and model complexity hinders the practical adoption of AI in these domains. Explainable AI (XAI) tackles this issue by attaching explanations to the models [3, 4, 5].

Among different explanations, feature attribution measures the contribution of input features to a model's prediction. In particular, local model-agnostic methods compute the input importance at instance level regardless of target model's architecture [6]. There are two major branches of local model-agnostic attribution: Locally Interpretable Model Explanation (LIME) [7] and game-theoretic techniques. On the one hand, LIME fits a surrogate model $g_\theta$ to randomly sampled perturbations around the target instance $x$ with a locality-defining kernel $\pi$. Most LIME-based technique use a linear $g_\theta$ since $\theta$ corresponds directly to importance, and most improvements are derived from modifying the noise generation process or the fitting process [8, 9, 10].

The second branch of local model-agnostic attribution is game-theoretic XAI. These methods approach the explanation process as a cooperative game, considering input features as players and the model as the value function. The game theory solution of a player's contribution corresponds directly to a feature's importance. The main strength of game-theoretic attribution is that they satisfy the underlying axioms of the corresponding solution. For example, the Shapley value [11]:

Submitted to 39th Conference on Neural Information Processing Systems (NeurIPS 2025). Do not distribute.

$$\phi_i = \frac{1}{n} \sum_{S \subseteq N \setminus i} \binom{n-1}{|S|} [v(S \cup i) - v(S)] \tag{1}$$

is a solution of cooperative game theory that uniquely satisfies linearity, dummy, symmetry, and efficiency. While the combinatorial nature of Shapley values make it impossible to calculate exactly for large number of features, KernelSHAP [12] shows that it can be approximated with a weighted linear regression. Due to its massive popularity, KernelSHAP has been explored thoroughly in the past literature [13, 14, 15, 16].

Unfortunately, KernelSHAP is suffers from issues like numerical instability. Consequently, more recent literature focuses on relaxing some axioms to improve the quality of the explanations. One example is the Banzhaf value [17], which is another solution of cooperative game theory which satisfies the same axioms as the Shapley value except efficiency:

$$\phi_i = \frac{1}{2^{n-1}} \sum_{S \subseteq N \setminus i} [v(S \cup i) - v(S)] \tag{2}$$

The Banzhaf value is simply an average of the payoff difference caused by player $i$ across all possible coalitions excluding said player. More generally, values of the form:

$$\phi_i = \sum_{S \subseteq N \setminus i} p(S)[v(S \cup i) - v(S)] \tag{3}$$

where $p(S)$ is the probability of coalition $S$, are referred to as probabilistic values [18].

One problem with regular Banzhaf value is that it assumes that all coalitions are equally likely to form. This assumption is equivalent assuming each player being neutral to joining a coalition. However, in real life, players are likely to have different preferences depending on their objectives. For example, if each player wishes to maximize their payoff, a player would have a higher probability of joining (i.e., a *preference*) the greater their expected payoff in larger coalitions. The criteria may not even be directly related to the game: for instance, if political parties vote on a regulation, they may make their vote not based on the game payoff (passing the regulation), but another criteria like future likelihood of re-election. Regardless of cause, reflecting the preference of coalition is critical for more accurate evaluation of each player's importance in a game.

Based on this notion, we introduce Preference Banzhaf, which computes Banzhaf value given each feature's probability of forming a coalition. We show that the attribution values can be computed through a centered (and later a regular) linear regression with binary masks, prove the convergence rate of the value, and empirically demonstrate the benefits of preference reflection. Our contributions are as follows:

- We introduce Preference Banzhaf, a novel algorithm that efficiently computes axiom-satisfying attribution using a different coalition-forming probability for each feature

- We show the equivalence between Preference Banzhaf and (a) a centered linear regression shifted by each feature's probability, and (b) a regular linear regression with intercept

- We derive the theoretical convergence rate of Preference Banzhaf

- We empirically demonstrate the effect of using Preference Banzhaf and interpret what the different weights mean intuitively

## 2 Related Work

### 2.1 Model-Agnostic Explanations

Model-agnostic explanations usually involve perturbing the input and measuring the change in the output. A fundamental method in this category is LIME [7], which fits an interpretable model with kernel-weighted loss. The original method uses a linear model with a radial basis function (RBF) kernel, but other kernels (such as cosine similarity kernel in Captum [19]) can be used.

Studies building upon LIME usually upgrade the sampling scheme or kernel selection. [9] trains a causal model for generating perturbations, and [20] uses a clustering model to select perturbations deterministically from the training dataset. [8] reformulates LIME as a Bayesian model to adjust the LIME coefficients by some prior. [21] adopts an empirical pipeline to measure optimal RBF kernel width for a desired level of local goodness of fit. [10] shows equivalence between RBF kernel and adjusted feature mask probability, significantly stabilizing the attribution results by removing the kernel from the regression.

## 2.2 Game-Theoretic XAI

Game theory-based XAI literature focuses on developing methods that satisfy certain axiomatic properties. They tend to use Shapley value [11] (Equation 1) as the basis, which satisfies four properties: linearity, dummy, symmetry, and efficiency. While the Shapley value is too costly to calculate exactly, [12] shows that it can be estimated using a linear regression, a method known as KernelSHAP. The method has been adapted in many different directions [16], such as architecture specialization [22, 23, 24] or estimation method improvements [25, 26]. One issue with Shapley value is that it can be numerically unstable and difficult to compute in practice. Recent works relax some of the axioms - mainly efficiency - to address these shortcomings. For example, [27] propose Beta Shapley, which adjust the Shapley averaging scheme to include a Beta distribution.

A growingly popular alternative is Banzhaf value (Equation 2). While similar in construction to Shapley value, they differ in the treatment of the order of feature subsets. For Shapley value, the order is important: a set of size $s$ that includes $i$ as the $m$-th element is different from that as the $l$-th element, assigning different weights to the two coalitions. Banzhaf value considers both sets to be the same and simply averages across all possible subsets. Despite this difference, the two values are extremely similar, especially in terms of the rank of contributions [28, 29].

Most papers that use Banzhaf value often use regular Banzhaf value. [29] uses Banzhaf value for data valuation; [30] uses Shapley and Banzhaf value to select the optimal vocabulary subset for NLP tasks; and [31] utilizes Banzhaf value to create counterfactuals in graph neural networks. [32] generalizes Banzhaf value to weighted Banzhaf value for data valuation and shows that optimal weight $w$ is dependent on the dataset and model. However, there has not been any research on computing Banzhaf values when all features have different weights, especially without relying on feature-wise calculations (referred to as Maximum Sample Reuse).

## 3 Method

### 3.1 Definition

Given players $i \in S \subseteq N$ , let $v(S)$ be the target value function for subset $S$. Let $w_i$ be the probability that player $i$ joins a coalition, i.e., their coalition *preference*. Then, the Preference Banzhaf value $\psi_p^i$ of player $i$ is defined as:

$$\psi_p^i = \sum_{S \in N \setminus i} [\prod_{j \in S} w_j \prod_{j \notin S} (1 - w_j)][v(S \cup i) - v(S)] \tag{4}$$

Intuitively, $\psi_p^i$ is the expected change in $v$ given that each player may join the coalition following a multivariate binomial distribution with parameter $\mathbf{w} = \{w_1, w_2, ..., w_d\}$. Regular Banzhaf value is a special case where $w_i = 0.5 \forall i$, while weighted Banzhaf value is another special case where $w_i = \alpha \forall i$.

### 3.2 Preference Banzhaf Approximation with Centered Linear Regression

KernelBanzhaf [33] approximates the Banzhaf value by masking each feature with probability $w = 0.5$, and regressing the results against $\mathbf{z} = \{-0.5, 0.5\}^d$, where $z_i = -0.5$ if $x_i$ is masked and 0.5 otherwise. This formulation can be generalized to any set of $w_i$ by using centered linear regression:

**Theorem 1. Preference Banzhaf as Centered Linear Regression.** *Preference Banzhaf $\psi_p$ is the solution of the centered linear regression:*

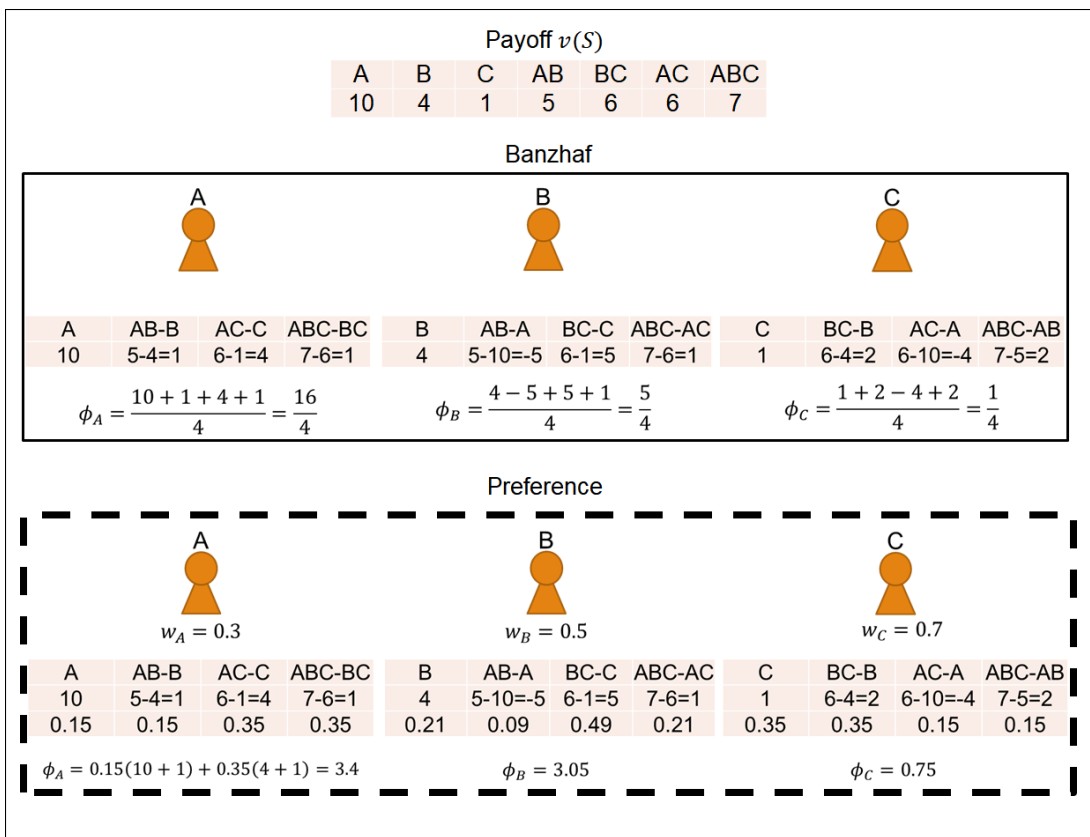

Figure 1: Illustration of regular versus Preference Banzhaf. Regular Banzhaf takes a simple average of payoff difference. Preference Banzhaf takes a weighted average of payoff difference based on the coalition-forming probability $w_i$.

$$\psi_p = arg\min_{\beta} E_X[(v(\mathbf{z}) - \beta^T\mathbf{z})^2] \tag{5}$$

where $z_i = m_i - w_i, p(m_i = 1) = w_i, m_i = 0, 1$.

We can further show that the solution is still $\psi_p$ after adding an intercept term.

**Theorem 2. Preference Banzhaf as Centered Linear Regression with Intercept.** *Preference Banzhaf $\psi_p$ is the solution of the centered linear regression with intercept:*

$$\beta_0^*, \psi_p = arg\min_{\beta_0,\beta} E_X[(v(\mathbf{z}) - \beta_0 - \beta^T\mathbf{z})^2] \tag{6}$$

The full proof for Theorems 1 and 2 are presented in the Appendix.

A consequence of Theorem 2 is that in terms of implementation, we do not need to center $\mathbf{z}$ to approximate the Preference Banzhaf value since centering does not affect the coefficients of a linear model when an intercept exists. We may perform the linear regression directly.

### 3.3 Convergence to True Value

A key question associated with kernel approximation of Banzhaf values is the rate of convergence to the true value. In the case of Preference Banzhaf value, it is closely related to GLIME [10] in implementation. Consequently, we can provide similar convergence guarantees.

**Theorem 3. Convergence of Preference Banzhaf** *Assume that $Z \sim \{b_i - w_i\}^d$, where $b_i \sim Ber(w_i)$. Then, given an empirical sample $Z_n$ and corresponding values $v_n$, the linear regression solution $\beta_n$ converges to $\psi_p$ with probability $1 - \delta$ (i.e., $P(|\beta_n - \psi_p|_2 \le \epsilon) \le 1 - \delta$*

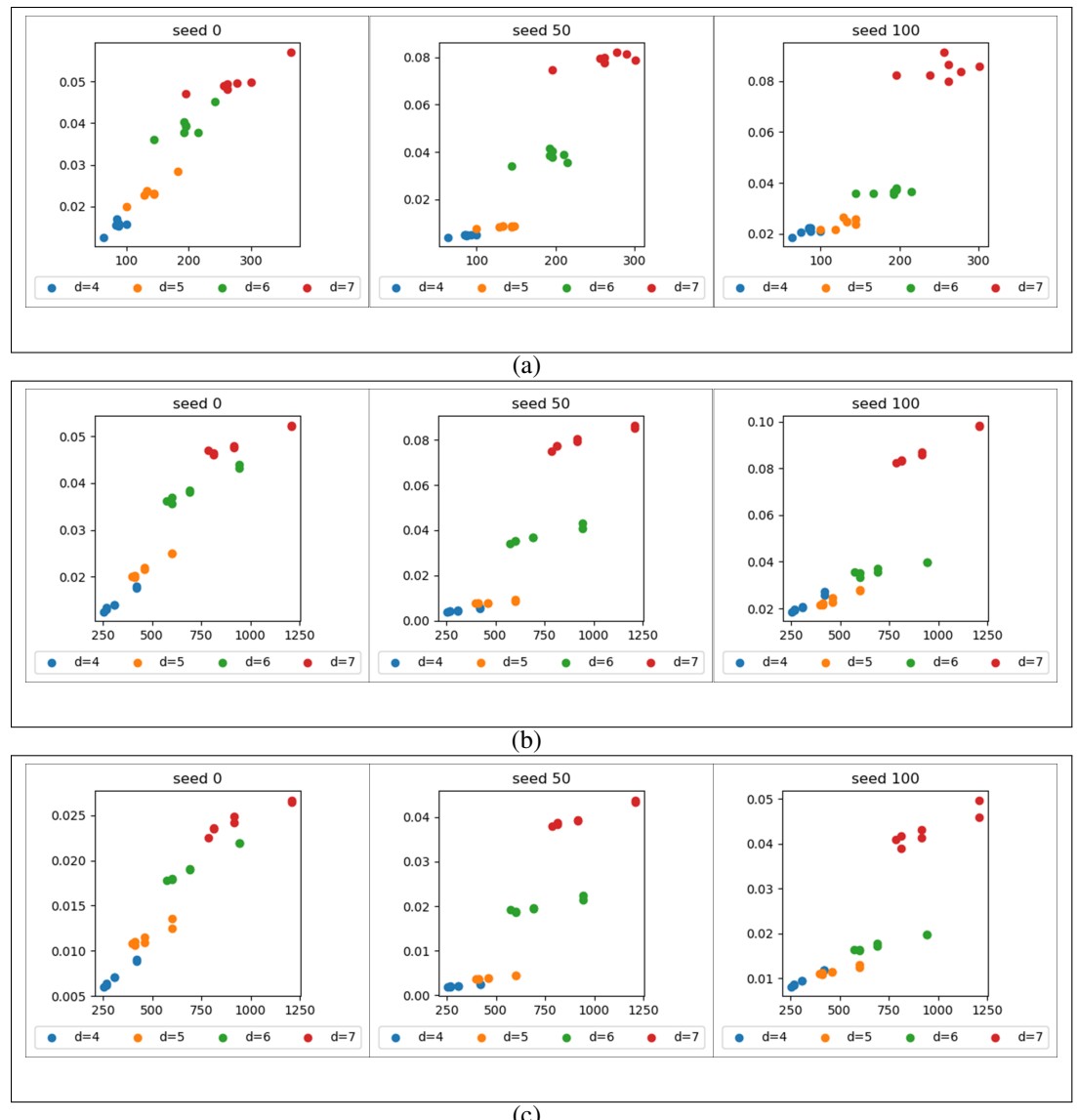

Figure 2: Convergence experiments. (a) $v^2\gamma^4$ for random probabilities and models generated from the seeds. (b) $\gamma^4$ when $v^2$ is constant at 0.25. (c) Same as (b) with $N = 2000$. Generally, $\gamma^4$ dominates the convergence relation with the volatility terms.

for $n = \Omega(\epsilon^{-2}M^2v^2d^3\gamma^4 log(4/\delta)))$ for some constant $M$, where $v^2 = max(w_i(1 - w_i))$ and $\gamma^2 = \sum_{i=1}^{d} 1/(w_i(1 - w_i))$.

The full proof for convergence is presented in the Appendix. This theorem implies that, with all else held constant, the solution converges the fastest when $w_i(1 - w_i)$ is maximized at $w_i = 0.5$, i.e., the regular Banzhaf value. It also implies that weighted Banzhaf values with $w_i = \alpha$ and $w_i = 1 - \alpha$ should have equal convergence under identical conditions.

## 3.4  Synthetic Experiment for Convergence

Figure 3.4 shows the plots $L_2$ error of Preference Banzhaf estimates against $v^2$ and $\gamma^4$ for synthetic datasets. Each subplot contains estimates for a model with 4 to 7 input features. The first row shows the relation between $L_2$ error and $v^2\gamma^4$ for random $w$ applied on random quadratic functions, while the second row shows the effect of $\gamma^4$ when $v^2$ is held constant at 0.25 (i.e., at least 1 $w_i$=0.5). We

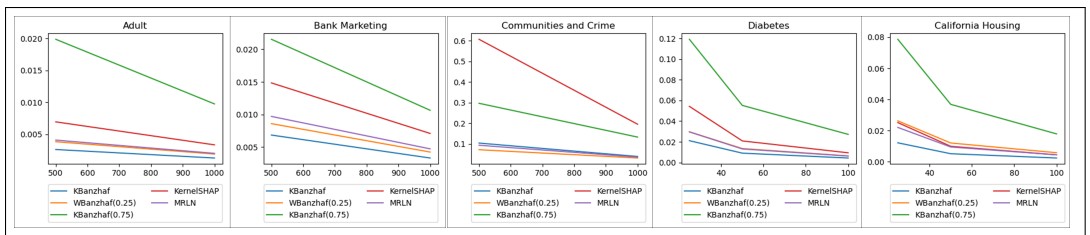

Figure 3: $L_2$-normalized error over $N$ across real datasets. We see that Kernel Banzhaf generally achieves the lowest sensitivity among Banzhaf methods as expected from Theorem 3.

see that the relation is linear for both cases. The third row shows the same plot as the second row except at $N = 2000$ instead of $N = 1000$. We see that while the maximum error decreases, the linear relation between error and $\gamma^4$ still holds. While not reported to conserve space, the relation between $L_2$ error and $v^2$ is generally constant or slightly linear, and $\gamma^4$ dominates most of the error relation.

## 4 Experiment

### 4.1 Setup

**Algorithms.** We use the following algorithms for the experiments:

- KernelBanzhaf ($Kbanzhaf$) [33]: this is equivalent to setting $w_i = 0.5$.

- Weighted Banzhaf with probability $\alpha$ ($WBanzhaf(\alpha)$): this is equivalent to setting $w_i = \alpha$. We use $\alpha$ of 0.25 and 0.75 to test the effect of $\alpha$ on convergence and explanation quality.

- KernelSHAP [12] ($KernelSHAP$): this method approximates Shapley value using linear regression with combinatorial kernel.

- $MRLN$ [34]: We use this method to choose $w_i$ for Preference Banzhaf. Model Response Localized Attribution (MRLN) computes the empirical probability by sorting the samples by a distance metric from the original instance and averaging the mask of the closest samples. We follow the original paper for the best empirical thresholds.

**Models and datasets.** We train an XGBoost classifier for several datasets (Adult Census, Communities and Crime, California Housing, Diabetes, and Bank Marketing). We use the default settings from training the classifiers. Each model is trained on an random 80% split of the corresponding dataset.

**Settings.** For the Adult and Diabetes datasets, which have only 8 features, we generate explanations with maximum sample size equal to $2^d$. For the rest of the tabular datasets, we use 500, 1000, and 2000 samples to evaluate the explanations. For tabular datasets, the evaluation is performed across 40 different seeds between 0 and 800. The replacement value for masking is a random instance in the opposite class. For image datasets, we use a baseline of 0 with a fixed seed of 0. The images are segmented into 64 equal segments. All evaluations are performed on the remaining 20% test split.

**Faithfulness.** We evaluate the faithfulness of the attributions using Area over Perturbation Curve [35] with predicted class's logit ($AOPC_L$) and probability ($AOPC_P$), as well as Iterative Removal of Features (IROF) [36]. It should be noted that while there are discussions on biases with these metrics [37, 38, 39], they are still widely used in the XAI literature for evaluation and it is outside of the scope of the study to discuss their limitations.

**Sensitivity.** We evaluate the sensitivity of the attributions using L2-normalized error [33], average pairwise rank correlation, and top-K Jaccard index. For The last metric, we set $K$ to 5 for Adult and Diabetes datasets, and the minimum between 20 and half of the number of features for the rest of the datasets. The sensitivity is evaluated only for tabular data due to computational constraints.

## 4.2 Quantitative Evaluation

### 4.2.1 Sensitivity

The $L_2$-normalized error for real datasets over $N$ is presented in Figure 4.2.1. It is immediately obvious that $Kbanzhaf$ achieves the lowest sensitivity amongst Banzhaf values, which agrees with Theorem 3 and the synthetic results since it minimizes $v^2\gamma^4$. $KernelSHAP$ achieves lower sensitivity than $WBanzhaf(0.75)$ in most datasets, but often loses to the other methods. $MRLN$ is surprisingly robust, achieving third or second lowest $L_2$-normalized error across all datasets. As will be shown in the subsequent section, $MRLN$ also achieves higher average faithfulness than other methods, which suggests that we may generate high fidelity explanations with small robustness tradeoff by adjusting $w_i$ to a model's internal behavior. The sensitivity measured using Jaccard distance and correlation index (reported in Appendix C) also agree with that using $L_2$-normalized error.

### 4.2.2 Faithfulness

The faithfulness evaluation of experiments on real tabular datasets is reported in Table 1. We can observe several patterns:

- Excluding Preference Banzhaf with $MRLN$ setup, the average faithfulness is generally the highest for regular Banzhaf value.

- The average standard error of faithfulness (the standard deviation of a metric for each instance divided by square root of number of seeds, averaged across instances) follows similar order as sensitivity: generally, $KBanzhaf$ is the smallest, followed by $WBanzhaf(0.25)$ or $MRLN$, then $KernelSHAP$ and $WBanzhaf(0.75)$.

- The average standard error of faithfulness for $WBanzhaf(0.75)$ tends to be much larger than the others. In particular, the average standard error for $MRLN$ is comparable to $WBanzhaf(0.25)$ despite the additional randomness caused by probability estimation.

These patterns demonstrate the effectiveness of using properly adjusted probabilities for Banzhaf values: we can achieve high and stable average fidelity.

## 4.3 Qualitative Evaluation

In this section, we analyze samples from image datasets to investigate the information captured by $w_i$. Specifically, we compare the faithfulness of explanations depending on the location of high $w_i$ with respect to the true object in the image. Given a 64 equally divided segmentation map, we use $w_i = 0.7$ (high weight) and $w_i = 0.3$ (low weight) and either place the higher weight in the center $4 \times 4$ segments ($Banzhaf(Center)$) or the remaining periphery segments ($Banzhaf(Periph)$). Comparing the faithfulness between the two setups, we find the following patterns:

- In terms of average faithfulness, $Banzhaf(Center)$ has much higher fidelity than $Banzhaf(Periph)$ as shown in Table 2. Given that many images in the Imagenette and Imagewoof datasets have their objects at the center of the image, this result implies that a higher overlap between the object and $w_i$ results in more faithful attributions.

- This pattern coincides with instance-level differences. In Figure 4.3, we have examples where $Banzhaf(Center)$ has much higher fidelity metric than $Banzhaf(Periph)$ and vice versa. We see that when faithfulness of $Banzhaf(Center)$ is higher, the main object is usually at the center. In the opposite case, the object is off-center or is too small compared to the window size.

This trend suggests that, to generate more faithful explanations, we need to select $w_i$ that is effectively the 'attention' of the model: higher $w_i$ should be assigned to features that the model focuses on for its predictions. It also explains why MRLN has higher average fidelity than other methods: it dynamically selects $w_i$ that aligns with the 'attention' of the model based on the target model's internal behavior. Note that the interpretation of $w_i$ is slightly different from an attribution, which determines how $much$ (in positive or negative direction) a segment contributes to a prediction. $w_i$ only implies that the segment is important - we do not know the $direction$ of said importance.

Table 1: Average Faithfulness and Standard Errors for Tabular Datasets

| Name | Logit_AOPC | Prob_AOPC | Logit_IROF |
|---|---|---|---|
| $WBanzhaf(0.25)$ | $1.3794 \pm 0.0041$ | $0.5551 \pm 0.0010$ | $0.3624 \pm 0.0011$ |
| $Kbanzhaf$ | $1.3866 \pm 0.0040$ | $0.5589 \pm 0.0009$ | $0.3578 \pm 0.0011$ |
| $WBanzhaf(0.75)$ | $1.2480 \pm 0.0084$ | $0.5259 \pm 0.0022$ | $0.3957 \pm 0.0024$ |
| $KernelSHAP$ | $1.3831 \pm 0.0052$ | $0.5572 \pm 0.0013$ | $0.3598 \pm 0.0015$ |
| $MRLN$ | $1.4271 \pm 0.0038$ | $0.5667 \pm 0.0009$ | $0.3486 \pm 0.0010$ |

(a) Bank Marketing

| Name | Logit_AOPC | Prob_AOPC | Logit_IROF |
|---|---|---|---|
| $WBanzhaf(0.25)$ | $5.5966 \pm 0.0090$ | $0.8703 \pm 0.0004$ | $0.0607 \pm 0.0004$ |
| $Kbanzhaf$ | $5.6177 \pm 0.0096$ | $0.8763 \pm 0.0004$ | $0.0544 \pm 0.0004$ |
| $WBanzhaf(0.75)$ | $5.1575 \pm 0.0261$ | $0.8666 \pm 0.0010$ | $0.0641 \pm 0.0010$ |
| $KernelSHAP$ | $5.3053 \pm 0.0226$ | $0.8562 \pm 0.0015$ | $0.0746 \pm 0.0015$ |
| $MRLN$ | $5.6942 \pm 0.0080$ | $0.8782 \pm 0.0003$ | $0.0525 \pm 0.0003$ |

(b) Communities and Crime

| Name | Logit_AOPC | Prob_AOPC | Logit_IROF |
|---|---|---|---|
| $WBanzhaf(0.25)$ | $2.7230 \pm 0.0025$ | $0.6417 \pm 0.0003$ | $0.2666 \pm 0.0004$ |
| $Kbanzhaf$ | $2.7221 \pm 0.0020$ | $0.6432 \pm 0.0003$ | $0.2648 \pm 0.0003$ |
| $WBanzhaf(0.75)$ | $2.6663 \pm 0.0059$ | $0.6370 \pm 0.0008$ | $0.2718 \pm 0.0008$ |
| $KernelSHAP$ | $2.7234 \pm 0.0031$ | $0.6424 \pm 0.0004$ | $0.2656 \pm 0.0005$ |
| $MRLN$ | $2.7396 \pm 0.0022$ | $0.6447 \pm 0.0003$ | $0.2630 \pm 0.0004$ |

(c) Adult

| Name | Logit_AOPC | Prob_AOPC | Logit_IROF |
|---|---|---|---|
| $WBanzhaf(0.25)$ | $3.4762 \pm 0.0082$ | $0.7881 \pm 0.0010$ | $0.1839 \pm 0.0011$ |
| $Kbanzhaf$ | $3.5125 \pm 0.0063$ | $0.7972 \pm 0.0005$ | $0.1745 \pm 0.0005$ |
| $WBanzhaf(0.75)$ | $3.4157 \pm 0.0168$ | $0.7964 \pm 0.0012$ | $0.1752 \pm 0.0012$ |
| $KernelSHAP$ | $3.4892 \pm 0.0095$ | $0.7919 \pm 0.0012$ | $0.1798 \pm 0.0012$ |
| $MRLN$ | $3.5303 \pm 0.0073$ | $0.7971 \pm 0.0006$ | $0.1746 \pm 0.0007$ |

(d) Diabetes

| Name | Logit_AOPC | Prob_AOPC | Logit_IROF |
|---|---|---|---|
| $WBanzhaf(0.25)$ | $4.4203 \pm 0.0068$ | $0.7983 \pm 0.0004$ | $0.1310 \pm 0.0005$ |
| $Kbanzhaf$ | $4.4382 \pm 0.0049$ | $0.8010 \pm 0.0002$ | $0.1280 \pm 0.0002$ |
| $WBanzhaf(0.75)$ | $4.3462 \pm 0.0154$ | $0.7990 \pm 0.0006$ | $0.1301 \pm 0.0006$ |
| $KernelSHAP$ | $4.4308 \pm 0.0069$ | $0.8002 \pm 0.0004$ | $0.1288 \pm 0.0004$ |
| $MRLN$ | $4.4485 \pm 0.0059$ | $0.8011 \pm 0.0003$ | $0.1279 \pm 0.0003$ |

(e) California Housing

Table 2: Average Faithfulness for High Probability at the Center and at the Periphery for Images

| Name | Logit_AOPC | Prob_AOPC | Logit_IROF |
|---|---|---|---|
| $Banzhaf(Center)$ | 5.0321 | 0.7224 | 0.2049 |
| $Banzhaf(Periph)$ | 4.8623 | 0.7154 | 0.2113 |

(a) Imagenette

| Name | Logit_AOPC | Prob_AOPC | Logit_IROF |
|---|---|---|---|
| $Banzhaf(Center)$ | 5.2139 | 0.7388 | 0.1413 |
| $Banzhaf(Periph)$ | 5.0047 | 0.7310 | 0.1482 |

(b) Imagewoof

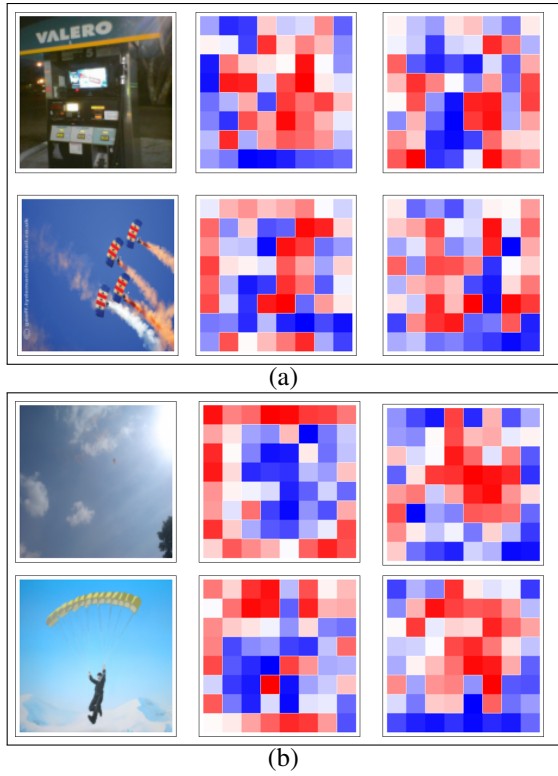

Figure 4: Examples (first column) from Imagenette and segment importance for (a) high positive faithfulness difference between $Banzhaf(Center)$ (2nd column) and $Banzhaf(Periph)$ (3rd column), and (b) high negative difference. The object tends to be large and at the center for the former, while it is small or off-center for the latter.

## 5  Conclusion

In this paper, we present Preference Banzhaf, where each input feature is masked following a different probability, i.e., their preference of forming a coalition. We prove that Preference Banzhaf values can be computed through (a) a centered linear regression without intercept, and (b) a regular linear regression with intercept. We also derive the theoretical convergence given a set of preferences. We compare the faithfulness and sensitivity of MLRN-based Preference Banzhaf against different model-agnostic baseline methods across several tabular and image datasets. We find that Preference Banzhaf achieves the best average fidelity across all datasets, often followed by vanilla Banzhaf values. In terms of sensitivity, vanilla Banzhaf achieves the lowest sensitivity across all datasets, but is usually closely followed by Preference Banzhaf.

## 6  Limitations and Future Directions

There are several limitations to this work. Firstly, this paper focuses on accurately computing Preference Banzhaf values given $w_i$. Discovering methods of finding optimal $w_i$ for a given objective using the relation between Preference Banzhaf and linear regression would be interesting. Secondly, Preference Banzhaf is limited to fixed $w_i$. Finding a fuzzy equivalent could help extend game-theoretic XAI to more diverse set of model-agnostic explanations. Lastly, this research focuses solely on feature attribution task. Extending Preference Banzhaf to other tasks such as data valuation could show the benefits of using more generalized forms of game-theoretic XAI in different applications.

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

# A  Experimental Details

The XGBoost classifiers are trained with default parameters from the xgboost package, while the image classifiers are fine-tuned from IMAGENET10K weight available in the torchvision package. The classification layer of the image classifiers consist of 4 linear layers with 20% dropout, batch normalization, and ReLU activation. All training and experiments are performed on Intel(R) Xeon(R) Gold 6342 CPU @ 2.8GHz and NVidia RTX A6000 (48GB).

Table 3: Model details.

| DATASET | MODEL | PACKAGE | ACC (%) |
|---------|-------|---------|---------|
| ADULT | XGBOOST | XGBOOST | 87.29 |
| CALIFORNIA | XGBOOST | XGBOOST | 84.74 |
| CRIME | XGBOOST | XGBOOST | 80.75 |
| IMAGENETTE | RESNET101 | TORCHVISION | 89.81 |
| IMAGEWOOF | RESNET101 | TORCHVISION | 79.89 |

Table 4: MLP layer details.

| BLOCK | LAYERS |
|-------|--------|
| 1 | RELU |
| 1 | LINEAR(2048,1024) |
| 1 | BATCHNORM |
| 2 | RELU |
| 2 | DROPOUT(0.2) |
| 2 | LINEAR(1024,512) |
| 2 | BATCHNORM |
| 3 | RELU |
| 3 | DROPOUT(0.2) |
| 3 | LINEAR(512,256) |
| 3 | BATCHNORM |
| 4 | RELU |
| 4 | DROPOUT(0.2) |
| 4 | LINEAR(512,10) |

## B    Proofs

360    In this section, we present the full proofs for theorems 1 through 3.

### B.1    Proof for Theorem 1

362    Expanding the objective, we have:

$$E[(f(x) - \beta^T x)^2] = E[(f(x) - \sum_{i=1}^{d} \beta_i x_i)^2] = E[f^2 - 2f \sum_{i=1}^{d} \beta_i x_i + \sum_{i=1}^{d} \sum_{j=1}^{d} \beta_i x_i \beta_j x_j]$$

$$= E[f^2 - 2f \sum_{i=1}^{d} \beta_i x_i + \sum_{i=1}^{d} \beta_i^2 x_i^2 + \sum_{i \neq j}^{d} \beta_i \beta_j x_i x_j]$$

$$= E[(1-d)f^2 + \sum_{i=1}^{d} (f - \beta_i x_i)^2 + \sum_{i \neq j}^{d} \beta_i \beta_j x_i x_j]$$

$$= (1-d)E[f^2] + \sum_{i=1}^{d} E[(f - \beta_i x_i)^2] + \sum_{i \neq j}^{d} \beta_i \beta_j E[x_i x_j] \tag{7}$$

363    $x_i$ are independent Bernouilli variables with probability $w_i$, which means $Cov(x_i, x_j) = 0$.
364    Therefore, if we center $x_i$ so that $E(x_i) = 0$, i.e., subtract $w_i$, then $E(x_i x_j) = Cov(x_i, x_j) +$
365    $E(x_i)E(x_j) = 0$.

366    Then, the equation changes to:

$$\beta_{pref} = arg \min_{\beta}[(1-d)E[f^2] + \sum_{i=1}^{d} E[(f - \beta_i x_i)^2]] = arg \min_{\beta}[\sum_{i=1}^{d} E[(f - \beta_i x_i)^2]] \tag{8}$$

367    which is equivalent to minimizing $\beta_{pref,i}$ individually. Taking the derivative for a single $\beta_{pref,i}$, we
368    have:

$$\frac{dE[(f - \beta_i x_i)^2]}{d\beta_i} = E[-2x_i(f - \beta_i x_i)] = 0$$

$$\rightarrow \beta_i = E[x_i f]/E[x_i^2] \tag{9}$$

369    Since $E[x_i^2] = Var(x_i) = w_i(1 - w_i)$ and $E[x_i f] = w_i(1 - w_i)E[f|x_i = 1 - w_i] + (1 -$
370    $w_i)(-w_i)E[f|x_i = -w_i]$:

$$\beta_i = \frac{w_i(1-w_i)E[f|x_i = 1 - w_i] + (1-w_i)(-w_i)E[f|x_i = -w_i]}{w_i(1-w_i)}$$

$$= E[f|x_i = 1 - w_i] - E[f|x_i = -w_i] \tag{10}$$

371    Since $x_i = 1 - w_i$ means feature $i$ is included in the input set $S$ and $x_i = -w_i$ means it is excluded
372    from $S$, the above equation becomes:

$$\beta_i = E[f(i \cup S)] - E[f(S)]$$

$$= \sum_{S \subseteq N \setminus i} [\prod_{j \in S} w_j \prod_{j \notin S} (1 - w_j)][f(S \cup i)] - \sum_{S \subseteq N \setminus i} [\prod_{j \in S} w_j \prod_{j \notin S} (1 - w_j)][f(S)]$$

$$= \sum_{S \subseteq N \setminus i} [\prod_{j \in S} w_j \prod_{j \notin S} (1 - w_j)][f(S \cup i) - f(S)] = \beta_{pref,i} \tag{11}$$

## B.2 Proof for Theorem 2

Equation 6 is identical to 5 except that we have the intercept term $\beta_0$. Expanding the equation, we have:

$$E[(f(x) - \beta_0 - \beta^T x)^2]$$
$$= E[(f(x) - \beta_0 - \sum_{i=1}^{d} \beta_i x_i)^2]$$
$$= E[f^2 - 2f\sum_{i=1}^{d}\beta_i x_i + \sum_{i=1}^{d}\sum_{j=1}^{d}\beta_i x_i \beta_j x_j + \beta_0^2 - 2\beta_0 f + 2\beta_0\sum_{i=1}^{d}\beta_i x_i]$$
$$= E[f^2 - 2f\sum_{i=1}^{d}\beta_i x_i + \sum_{i=1}^{d}\beta_i^2 x_i^2 + \sum_{i\neq j}^{d}\beta_i\beta_j x_i x_j + \beta_0^2 - 2\beta_0 f + 2\beta_0\sum_{i=1}^{d}\beta_i x_i]$$
$$= E[(1-d)f^2 + \sum_{i=1}^{d}(f - \beta_i x_i)^2 + \sum_{i\neq j}^{d}\beta_i\beta_j x_i x_j + \beta_0^2 - 2\beta_0 f + 2\beta_0\sum_{i=1}^{d}\beta_i x_i]$$
$$= (1-d)E[f^2] + \sum_{i=1}^{d}E[(f - \beta_i x_i)^2] + \sum_{i\neq j}^{d}\beta_i\beta_j E[x_i x_j] + \beta_0^2 - 2\beta_0 E[f] + 2\beta_0\sum_{i=1}^{d}\beta_i E[x_i] \tag{12}$$

Since centering sets $E[x_i] = 0$ and $E[x_i x_j] = 0$:

$$\beta_{pref} = arg\min_{\beta}[(1-d)E[f^2] + \sum_{i=1}^{d}E[(f - \beta_i x_i)^2] + \beta_0^2 - 2\beta_0 E[f]] = arg\min_{\beta}[\sum_{i=1}^{d}E[(f - \beta_i x_i)^2]] \tag{13}$$

Since the objective is equivalent, the solution stays identical as that from Equation 5.

## B.3 Proof for Theorem 3

This proof closely follows the convergence of GLIME [10]. Since Preference Banzhaf is the solution for a linear regression model, we know that:

$$\phi_{pref} = (X_n^T X_n)^{-1} X_n y_n \tag{14}$$

where $X_n$ is the centered sampled masks and $y_n$ is the corresponding model predictions. Representing $\Sigma_n = X_n^T X_n$ and $\Gamma_n = X_n y_n$, we would like to find the convergence of $\Sigma_n^{-1}\Gamma_n$ to the limit $\Sigma^{-1}\Gamma$. First, we can find the limit for $\Sigma_n$ as:

$$\Sigma = \lim_{n\to\infty}\Sigma_n = \lim_{n\to\infty}X_n^T X_n = E(X^T X) = Var(X) = diag(\sigma_i^2) = diag(w_i(1-w_i)) \tag{15}$$

$E(X^T X)$ is equal to the variance of $X$ since $X$ has been centered, i.e., $E(x_i) = 0 \forall i$, which makes $Cov(x_i, x_j) = E(x_i x_j) - E(x_i)E(x_j) = E(x_i x_j)$. Note that $0 \leq \sigma_i^2 \leq 0.25$ since each mask follows a Bernouilli distribution. We can also bound the values of $\Sigma_n$ as follows:

$$\hat{\sigma_n^i} = \frac{1}{n}\{\sum_{k\in S_1} w_i^2 + \sum_{k\in S_2}(1-w_i)^2\} \leq \frac{1}{n}\sum_{k=1}^{n} max(w_i, 1-w_i)^2 \tag{16}$$

$$\hat{\sigma_n}^{ij} = \frac{1}{n}\{\sum_{k \in S_1} w_i w_j + \sum_{k \in S_2} -w_i(1 - w_j)$$

$$+ \sum_{k \in S_3} -(1 - w_i)w_j + \frac{1}{n}\sum_{k \in S_4} (1 - w_i)(1 - w_j)\} \tag{17}$$

$$\leq \frac{1}{n}\sum_{k=1}^{n} max(w_i w_j, (1 - w_i)(1 - w_j) \leq 1$$

Therefore, all elements of $||\Sigma_n - \Sigma||$ are bounded to $[-0.25, 1]$, and we may apply matrix Hoeffding's inequality with $v^2 = max(\sigma_i^2)$:

$$P(||\Sigma_n - \Sigma||_2 \geq t) \leq 2dexp\left(-\frac{nt^2}{8v^2}\right) \tag{18}$$

$||\Sigma^{-1}||_F^2$ is simply the sum of inverse of variances $\sum_d 1/\sigma_i^2 = \gamma^2$. Lastly, we may apply Hoeffding's inequality to $\Gamma_n$ to find:

$$P(||\Gamma_n - \Gamma||_2 \geq t) \leq 2dexp\left(-\frac{nt^2}{8M^2d^2}\right) \tag{19}$$

Following [10], if we let $n$ be the maximum among $n_1 = 32\gamma^2 v^2 log(4d/\delta)$, $n_2 = 32\epsilon^{-}2M^2 d^2 \gamma^2 log(4d/\delta)$, and $n_3 = 32\epsilon^{-}2M^2 v^2 d\gamma^4 log(4d/\delta)$, we have $P(||\Sigma_n^{-1}\Gamma_n - \Sigma^{-1}\Gamma|| \leq 1 - \delta)$.

## C   Sensitivity

The sensitivity results using Jaccard distance and correlation index are as follows. The results agree with that in the main figure with $L_2$-normalized error.

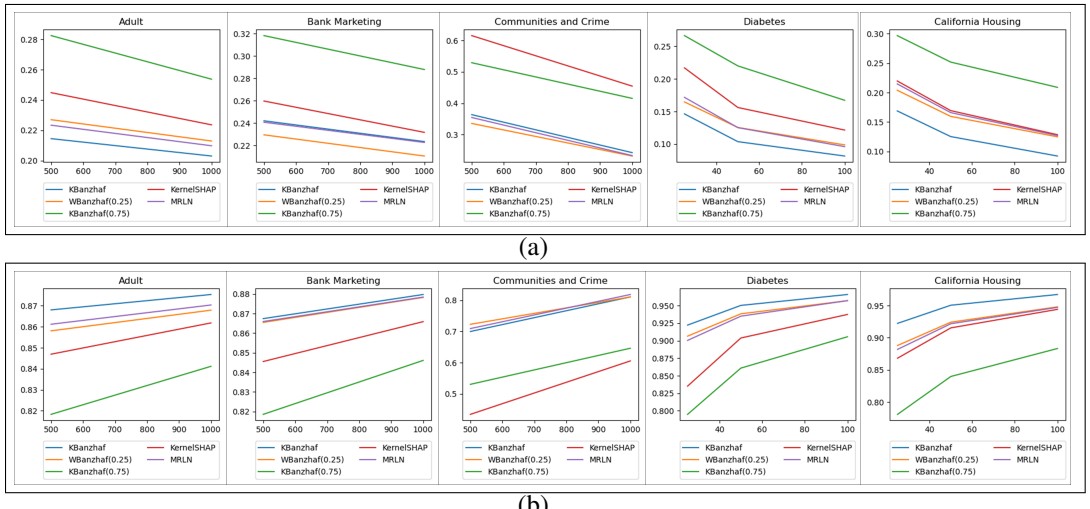

Figure 5: (a) Jaccard distance and (b) correlation index across different datasets. The patterns match those implied by $L_2$-normalized error in Figure 4.2.1.

