# OpenReview forum: "Preference Banzhaf: A Game-Theoretic Index with Feature-wise Probabilities"
_NeurIPS.cc/2025/Conference — Submitted to NeurIPS 2025_

### Official Review · Reviewer_vTBM · 2025-06-16

**Clarity:** 2
**Significance:** 2
**Originality:** 2
**Rating:** 2
**Confidence:** 4

**Summary:**

This paper advocated for weighted Banzhaf values, where each player $i \in N$ joins a coalition with probability $w_i$. The theoretical analysis of these values has been carried out in [A], where this paper reproduces some of the results in Theorem 1 and 2. The paper proposes "Preference Banzhaf" as a family of weighted Banzhaf values, which are determined by finding the "preferences of joining a coalition" $w_i$. The empirical section evaluates these weighted Banzhaf values with $w_i \equiv 0.75, w_i \equiv 0.25$, and $w_i \equiv 0.5$, as well as weights $w_i$ found by a "MRLN", a method not further described nor referenced. The empirical analysis compares these weighted Banzhaf values with approximated Shapley values from KernelSHAP on an XGBoost model and on ResNet101 using faithfulness metrics, such as perturbation curves and iterativel removal of features. The methods show comparable performance with the Shapley value  and KernelSHAP on several benchmark datasets.  MRLN achieves mostly superior results. Moreover, "sensitivity" is measured as well as qualitative examples in a CV setting.

**Questions:**

- Is it reasonable to choose independent probabilities per player? What is the interpretation of these probabilities? I was unable to follow how you identify these probabilities and what are their interpretation? Could you describe MRLN?
- Could you provide more insights in to the computed metrics and their purpose, possibly in the appendix?

**Ethical Concerns:**

["NO or VERY MINOR ethics concerns only"]

**Final Justification:**

I decided to keep my score, as in my view the paper in its current form is not ready for acceptance. The theoretical contribution is strongly connected to existing literature [A] and the empirical section is not sufficiently covered. I would also suggest for future revisions to put a stronger focus on the choice of weights and their implications for practitioners, as this is the (novel) XAI-specific part of this work.

**Limitations:**

The authors discuss limitations of their work.

**Paper Formatting Concerns:**

Minor concerns regarding figure references were stated above.

**Quality:**

1

**Strengths And Weaknesses:**

This paper seems to be in preliminary stage with substantial flaws. The theoretical results have limited novelty, the experimental details are unclear in several aspects, such as implementation details and evaluation. While applying weighted Banzhaf values complementary to Shapley values could be an interesting direction of research, the benefits and design choices, e.g. finding the probabilities $w_i$, are not well explored.

**Strengths**
- The paper proposes an interesting concept to include the probability of a player joining the coalition in the computation of game-theoretic attribution scores.

**Weaknesses**
- The theoretical results, especially Theorem 1 and Theorem 2 are mostly known. Theorem 1 and the weighted least-squares representation of weighted Banzhaf values is well known, and has been proven, e.g. in [A]. Moreover, Theorem 2 follows from the dummy axiom, i.e. constant shifts in the value function do not affect the weighted Banzhaf values.
- It remains unclear what the interpretation of the probabilities $w_i$ would be in practice. I was unable to find reference [34] online, which seems to be at the core of finding these probabilities with the MRLN method. Moreover, a key limitation of the method is that the probabilities are chosen independently per feature/player, which might be unreasonable.
- The experimental section is quite chaotic. Additional insights into the computed metrics or additional plots, e.g. for AOPC, IROF metrics, would be helpful in the appendix. In the sensitivity plots, I would expect standard deviations and I was unable to follow what kind of sensitivity is measured here. The improvements seem overall quite minor, and implementation details are unclear, e.g. for MRLN. It is also unclear to me what should be observed from Figure 4.

**Minor**
-  Figures are generally wrongly referenced, e.g. "Figure 4.2.1" in line 181, "Figure 3.4" in line 141, "Figure 4.3" in line 216

[A] Marichal, Jean-Luc, and Pierre Mathonet. "Weighted Banzhaf power and interaction indexes through weighted approximations of games." European journal of operational research 211.2 (2011): 352-358.

---

> ### Author Rebuttal · Authors · 2025-07-30
>
> Thank you very much for your review. Our responses are as follows:
>
> **C1: The theoretical results, especially Theorem 1 and Theorem 2 are mostly known. Theorem 1 and the weighted least-squares representation of weighted Banzhaf values is well known, and has been proven, e.g. in [A]. Moreover, Theorem 2 follows from the dummy axiom, i.e. constant shifts in the value function do not affect the weighted Banzhaf values.**
>
> A: We were unaware of the proof and have added it as a reference. For Theorem 2, while it does follow from the dummy axiom, it should be considered separate from Theorem 1 because the presence of intercept changes sensitivity in terms of implementation. Typical game-theoretic formulation that follows Theorem 1 does not include an intercept (such as KernelBanzhaf). In this case, the value is affected by vertical shifts since the intercept is forced to be 0, and is also not equivalent to uncentered linear regression.
>
> **C2: It remains unclear what the interpretation of the probabilities would be in practice. I was unable to find reference [34] online, which seems to be at the core of finding these probabilities with the MRLN method. Moreover, a key limitation of the method is that the probabilities are chosen independently per feature/player, which might be unreasonable.**
>
> A: In practice, the probabilities would be adjusted to user’s needs. For example, it is clear from convergence that we should use regular Banzhaf if we want to achieve best stability.
>
> In terms of intuition $w_i$ indicate 'where the user would like to focus on when generating the explanations to meet their objectives'. For example, we can sample specific coalition structures more frequently by matching the probabilities between features in the structures. Consider a model with 6 features, and we wish to select coalition structure [1,2,3] and [4,5,6] more frequently. If we set $w_{[1,2,3]}$ to 0.2 and [3,4,5] to 0.8, the sampled coalitions are more likely to select the two coalitions separately ($0.8^6$) than together ($0.2^3*0.8^3$).
>
> Another example is reflecting domain knowledge in XAI process. If a user considers or knows feature A to be important by domain knowledge, they may mask it more frequently so that the ranking of other features (after excluding the important feature) can be captured more correctly.
>
> The original motivation can be interpreted as coalition selection that maximizes total expected payoff (sum of Banzhaf values).
> MRLN is an example for selecting $w_i$ heuristically, and demonstrates that even heuristic adjustments can make improvements that achieve user objective. Furthermore, the goal of this paper is to show that we should choose $w_i$ flexibly to user goals, not how we should calculate them. We already discussed in the limitations that it would be worthwhile to find ways to select $w_i$ optimally for different objectives.
>
> **C3: The experimental section is quite chaotic. Additional insights into the computed metrics or additional plots, e.g. for AOPC, IROF metrics, would be helpful in the appendix. In the sensitivity plots, I would expect standard deviations and I was unable to follow what kind of sensitivity is measured here. The improvements seem overall quite minor, and implementation details are unclear, e.g. for MRLN. It is also unclear to me what should be observed from Figure 4.**
>
> **Q2: Could you provide more insights in to the computed metrics and their purpose, possibly in the appendix?**
>
> A:
>
> 1. We do not include standard deviations as a sensitivity measure since we are measuring the total error for the indices of all features, not for individual features. If you mean the standard deviation for sensitivities themselves, sensitivity measures are computed as an aggregate of multiple runs and thus does not have a standard deviation.
>
> 2. For the results, I initially thought it would be preferable to show raw values as tables over visualization since it gives access to the real values. However, it seems to reduce legibility of the results. I will change them to figures for better legibility and move the tables to the appendix.
>
> 3. While the gains seem minor due to their relative magnitudes, it is statistically significant based on difference-of-mean tests.
>
> 4. The main idea of MRLN is to (a) compute the model behavior (either as output or latent vector) from a uniform perturbation; (b) select outputs satisfying locality criteria in the behavior space; (c) compute empirical probability based on the masks of the selected outputs; and (d) perform linear regression with the new probability. This method heuristically aligns mask probability with model’s empirical behavior, making it suitable as an example for selecting masking probabilities when a user’s objective is to improve explanation fidelity.
>
> 5. Figure 4 was meant to show that the importance ranks of segments change significantly based on object’s location when we give low probability on the center or the periphery. In a sense, we are inserting ‘domain knowledge’ regarding the location of the object by adjusting the probabilities.
>
> 6. Regarding the metrics, the quantities are traditionally used metrics for measuring fidelity and sensitivity, so their descriptions had been removed due to text limit. I included the metric definitions in the appendix for clarification as follows (the paper itself could not be updated per NeurIPS regulations).
>
> In XAI, we typically want the explanations to satisfy several desiderata. Two typical properties discussed in the literature are faithfulness and sensitivity. Faithfulness (also known as fidelity) is the degree to which an explanation matches the model output behavior. Ideally, more important features would have greater impact on model decisions. Sensitivity (also known as robustness or consistency) measures the stability of the attribution for identical inputs. A good explanation should be stable, i.e., have lower sensitivity.
>
> In this paper, we considered Area over Perturbation Curve (AOPC) and Iterative Removal of Features (IROF) for faithfulness. AOPC measures the area over the curve (AOC) of the change in model output from the original as we replace each feature in the input with the chosen baseline:
>
> $$
> AOPC = \frac{1}{L+1}\sum^{L}_{k=1}{(f(x^0_M)-f(x^k_M))}
> $$
>
> Where $M$ is the feature replacement procedure and $L$ is the number of replacements. For classifiers, $f$ is usually the logit or probability of the target class. IROF follows a similar process, except that it works exclusively with probabilities:
>
> $$
> IROF = \frac{1}{L+1}\sum^{L}_{k=1}{(p(x^k_M)/p(x^0_M))}
> $$
>
> A model is more faithful if it has higher AOPC and lower IROF.
>
> For sensitivity, we evaluate $L_2$-normalized error between attributions across several runs and the true value. If the true value is difficult to calculate, we use the average as the proxy.
>
> $$
> err_{l2} = \frac{1}{|\bar{\beta}|^2_2}\sum^{L}_{k=1}|\beta_k-\bar{\beta}|_2^2
> $$
>
> We also use average pairwise rank correlation and top-K Jaccard distance amongst attributions.
>
> $$
> \rho = \frac{1}{L-1}\sum^{L-1}_{k=1}{corr( \beta_k, \beta_k + 1 )}
> $$
>
> and
>
> $$
> J_K = \frac{1}{L-1}\sum^{L-1}_{k=1}{1-\frac{|topk(\beta_k)\cap topk(\beta_k+1)|}{|topk(\beta_k)\cup topk(\beta_k+1)|}}
> $$
>
> where $L$ is the number of times the explanation has been evaluated. A method is less sensitive if it has lower $L_2$-normalized error, higher correlation, and higher Jaccard index.
>
>
> **M1: Figures are generally wrongly referenced, e.g. "Figure 4.2.1" in line 181, "Figure 3.4" in line 141, "Figure 4.3" in line 216**
>
> A: The error has been corrected.
>
> **Q1: Is it reasonable to choose independent probabilities per player? What is the interpretation of these probabilities? I was unable to follow how you identify these probabilities and what are their interpretation? Could you describe MRLN?**
>
> A:
>
> 1. Game theory-wise, it is definitely reasonable to assume that a player will try to maximize their payoff in a given game. Therefore, it is also reasonable that they would adjust their coalition-forming probabilities to meet this goal.
>
> In terms of feature attribution, it is still reasonable to choose probabilities individually to suite the user’s needs as mentioned above. In terms of intuition, $w_i$ indicate 'where the user would like to focus on when generating the explanations to meet their objectives'. For example, consider reflecting domain knowledge in XAI process. If a user considers or knows feature A to be important by domain knowledge, they may mask it more frequently so that the ranking of other features (after excluding the important feature) can be captured more correctly. The original motivation can be interpreted as coalition selection that maximizes total expected payoff (sum of Banzhaf values). In both cases, we are adjusting $w_i$ to user’s objectives.
>
> 2. Reiterating above comment, the main idea of MRLN is to (a) compute the model behavior (either as output or latent vector) from a uniform perturbation; (b) select outputs satisfying locality criteria in the behavior space; (c) compute empirical probability based on the masks of the selected outputs; and (d) perform linear regression with the new probability. This method heuristically aligns mask probability with model’s empirical behavior, making it suitable as an example for selecting masking probabilities. However, it should be emphasized that it is not the main objective of this paper to show MRLN is better, but that we can improve average fidelity through feature-wise probability selection.

---

> > ### Comment · Reviewer_vTBM · 2025-08-05
> >
> > I thank the authors for their detailed response and answering my question. I better understand the purpose of the weights now, as a user-specific choice to (partially) include or exclude specific features in the perturbations.
> > - I would have expected the standard deviation across multiple runs
> >
> > Overall I decided to keep my score, as in my view the paper in its current form is not ready for acceptance. The theoretical contribution is strongly connected to existing literature [A] and the empirical section is not sufficiently covered. I would also suggest for future revisions to put a stronger focus on the choice of weights and their implications for practitioners, as this is the (novel) XAI-specific part of this work.

---

### Official Review · Reviewer_iu3B · 2025-07-02

**Clarity:** 2
**Significance:** 2
**Originality:** 3
**Rating:** 4
**Confidence:** 4

**Summary:**

The paper introduces Preference Banzhaf, a novel game-theoretic feature attribution method that addresses limitations of existing approaches by relaxing the assumption that all feature coalitions are equally likely, and theoretically analyze the convergence properties of Preference Banzhaf. Extensive experiments on tabular and image datasets demonstrate the method's effectiveness.

**Questions:**

- Could you explicitly justify why feature-specific coalition probabilities are meaningful in ML explanations? A direct connection to model properties (e.g., feature redundancy, model architecture, or data distribution) would strengthen the case for Preference Banzhaf.
- Could you discuss limitations and potential pitfalls of using MRLN or similar heuristics for w_i estimation? Maybe you can include ablation studies on the impact of different w_i settings (e.g., random vs. learned) that would help validate the robustness of the method.
- Since you have mentioned recent advances of SHAP variants like Beta Shapley, could you include it as a baseline to compare to?

**Ethical Concerns:**

["NO or VERY MINOR ethics concerns only"]

**Final Justification:**

I read the rebuttals and the comments of other reviewers. I decrease the score of Clarity from 3 to 2.

**Limitations:**

Yes

**Paper Formatting Concerns:**

There are no formatting concerns.

**Quality:**

3

**Strengths And Weaknesses:**

Strengths:
- The proposed Preference Banzhaf addresses the limitation of uniform coalition assumptions in traditional methods.
- The paper establishes a theoretical equivalence between Preference Banzhaf and centered/regular linear regression, providing a fresh perspective on feature attribution.
- Extensive experiments on tabular and image datasets demonstrate the method's effectiveness.

Weaknesses:
- The paper motivates Preference Banzhaf by drawing an analogy to real-world games where players have preferences for coalition formation. However, this analogy is not clearly justified in the context of machine learning. In ML, features are not autonomous agents with preferences; their "coalition probabilities" (i.e., w_i) lack a principled interpretation. The paper does not convincingly address why such feature-specific probabilities should exist or how they align with the goals of model explanation. This raises questions about whether the problem being solved is truly relevant to XAI.
- A critical challenge in applying Preference Banzhaf is the difficulty of obtaining valid w_i values for real-world models. While the paper proposes using MRLN to estimate w_i, this approach relies on heuristics and does not provide theoretical guarantees for its validity. The experiments show improved performance when w_i is manually set to focus on central image regions (Banzhaf(Center)), but this scenario is contrived and does not generalize to arbitrary models or datasets. The paper lacks a systematic discussion of how to reliably estimate w_i in practice, especially for high-dimensional or black-box models where feature interactions are complex.
- In experiments, the compared baselines do not include recent advances in XAI (e.g., SHAP variants like Beta Shapley [1]). A more comprehensive comparison will strengthen the claims.

[1] Kwon Y, Zou J. Beta shapley: a unified and noise-reduced data valuation framework for machine learning [J]. arXiv preprint arXiv:2110.14049, 2021.

---

> ### Author Rebuttal · Authors · 2025-07-30
>
> Thank you very much for your review. Our responses to the comments and questions are as follows:
>
> **C1: The paper motivates Preference Banzhaf by drawing an analogy to real-world games where players have preferences for coalition formation. However, this analogy is not clearly justified in the context of machine learning. In ML, features are not autonomous agents with preferences; their "coalition probabilities" (i.e., w_i) lack a principled interpretation. The paper does not convincingly address why such feature-specific probabilities should exist or how they align with the goals of model explanation. This raises questions about whether the problem being solved is truly relevant to XAI.**
>
> **Q1: Could you explicitly justify why feature-specific coalition probabilities are meaningful in ML explanations? A direct connection to model properties (e.g., feature redundancy, model architecture, or data distribution) would strengthen the case for Preference Banzhaf.**
>
> A: The idea of choosing feature-specific probabilities is a natural extension of choosing different global masking probabilities like Weighted Banzhaf value and Beta Shapley value: the difference is that we propose further refining the probabilities to be individualized by instance and by input feature.
>
> For example, we can sample specific coalition structures more frequently by matching the probabilities between features in the structures. Consider a model with 6 features, and we wish to select coalition structure [1,2,3] and [4,5,6] more frequently. If we set $w_{[1,2,3]}$ to 0.2 and [3,4,5] to 0.8, the sampled coalitions are more likely to select the two coalitions separately ($0.8^6$) than together ($0.2^3*0.8^3$).
>
> Another example is reflecting domain knowledge in XAI process. If a user considers or knows feature A to be important by domain knowledge, they may mask it more frequently so that the ranking of other features (after excluding the important feature) can be captured more correctly.
>
> The original motivation can be interpreted as coalition selection that maximizes total expected payoff (sum of Banzhaf values).
>
> **C2: A critical challenge in applying Preference Banzhaf is the difficulty of obtaining valid w_i values for real-world models. While the paper proposes using MRLN to estimate w_i, this approach relies on heuristics and does not provide theoretical guarantees for its validity. The experiments show improved performance when w_i is manually set to focus on central image regions (Banzhaf(Center)), but this scenario is contrived and does not generalize to arbitrary models or datasets. The paper lacks a systematic discussion of how to reliably estimate w_i in practice, especially for high-dimensional or black-box models where feature interactions are complex.**
>
> A: We already discussed this point in the limitations. The goal of this paper is to show that $w_i$ should be chosen flexibly depending on the user's objective, and that the choice can improve explanation performance in terms of different criteria. In fact, MRLN’s heuristic nature, shows that it is desirable to choose probabilities by feature since even suboptimally-selected adjustments can make statistically significant gains in fidelity (which is the objective of MRLN). If the user’s goal was not high fidelity but high robustness, they would select different $w_i$.
>
> The qualitative experiment visually demonstrates this idea by showing that attributions rank the object-including segments more highly when $w_i$ is lower in the important region.
>
> **C3: In experiments, the compared baselines do not include recent advances in XAI (e.g., SHAP variants like Beta Shapley [1]). A more comprehensive comparison will strengthen the claims.**
>
> **[1] Kwon Y, Zou J. Beta shapley: a unified and noise-reduced data valuation framework for machine learning [J]. arXiv preprint arXiv:2110.14049, 2021.**
>
> **Q3: Since you have mentioned recent advances of SHAP variants like Beta Shapley, could you include it as a baseline to compare to?**
>
> A: We performed additional tests by implementing Beta Shapley and random weights (Q2). The faithfulness results are as follows. Qualitatively, the results stay consistent as the additional baselines do not perform better than the other methods. The results for sensitivity also stay consistent.
>
> Bank Marketing:
>
> | Name           | Logit AOPC      | Probability AOPC | IROF            |
> |----------------|-----------------|------------------|-----------------|
> | WBanzhaf(0.25) | 1.3794 ± 0.0041 | 0.5551 ± 0.001   | 0.3624 ± 0.0011 |
> | KBanzhaf       | 1.3866 ± 0.004  | 0.5589 ± 0.0009  | 0.3578 ± 0.0011 |
> | WBanzhaf(0.75) | 1.2479 ± 0.0084 | 0.5259 ± 0.0022  | 0.3957 ± 0.0024 |
> | KSHAP          | 1.3831 ± 0.0052 | 0.5572 ± 0.0013  | 0.3598 ± 0.0015 |
> | BetaShap(16,1) | 1.2946 ± 0.008  | 0.5317 ± 0.0029  | 0.3896 ± 0.0033 |
> | BetaShap(4,1)  | 1.3474 ± 0.0076 | 0.5459 ± 0.0022  | 0.373 ± 0.0026  |
> | BetaShap(1,4)  | 1.2433 ± 0.0125 | 0.5244 ± 0.0034  | 0.3984 ± 0.0039 |
> | BetaShap(1,16) | 1.032 ± 0.0144  | 0.4615 ± 0.0045  | 0.4686 ± 0.0051 |
> | Random         | 1.3074 ± 0.0223 | 0.5366 ± 0.0057  | 0.3834 ± 0.0066 |
> | PBanzhaf(MRLN) | 1.4271 ± 0.0038 | 0.5667 ± 0.0009  | 0.3486 ± 0.001  |
>
> Communities and Crime:
>
> | Name           | Logit AOPC      | Probability AOPC | IROF            |
> |----------------|-----------------|------------------|-----------------|
> | WBanzhaf(0.25) | 5.5966 ± 0.009  | 0.8703 ± 0.0004  | 0.0607 ± 0.0004 |
> | KBanzhaf       | 5.6177 ± 0.0096 | 0.8763 ± 0.0004  | 0.0544 ± 0.0004 |
> | WBanzhaf(0.75) | 5.1575 ± 0.0261 | 0.8666 ± 0.001   | 0.0641 ± 0.001  |
> | KSHAP          | 5.3053 ± 0.0226 | 0.8562 ± 0.0015  | 0.0746 ± 0.0015 |
> | BetaShap(16,1) | 5.0147 ± 0.0423 | 0.8223 ± 0.0042  | 0.1104 ± 0.0044 |
> | BetaShap(4,1)  | 5.3727 ± 0.0278 | 0.8508 ± 0.0022  | 0.0807 ± 0.0023 |
> | BetaShap(1,4)  | 5.4124 ± 0.029  | 0.8669 ± 0.0014  | 0.0639 ± 0.0014 |
> | BetaShap(1,16) | 4.8278 ± 0.0502 | 0.8471 ± 0.0027  | 0.0845 ± 0.0028 |
> | Random         | 5.4366 ± 0.0288 | 0.8699 ± 0.0012  | 0.061 ± 0.0013  |
> | PBanzhaf(MRLN) | 5.6942 ± 0.008  | 0.8782 ± 0.0003  | 0.0525 ± 0.0003 |
>
> Adult:
>
> | Name           | Logit AOPC      | Probability AOPC | IROF            |
> |----------------|-----------------|------------------|-----------------|
> | WBanzhaf(0.25) | 2.723 ± 0.0025  | 0.6417 ± 0.0003  | 0.2666 ± 0.0004 |
> | KBanzhaf       | 2.7221 ± 0.002  | 0.6432 ± 0.0003  | 0.2648 ± 0.0003 |
> | WBanzhaf(0.75) | 2.6663 ± 0.0059 | 0.637 ± 0.0008   | 0.2718 ± 0.0008 |
> | KSHAP          | 2.7234 ± 0.0031 | 0.6424 ± 0.0004  | 0.2656 ± 0.0005 |
> | BetaShap(16,1) | 2.68 ± 0.0062   | 0.6329 ± 0.0015  | 0.2766 ± 0.0017 |
> | BetaShap(4,1)  | 2.7095 ± 0.0048 | 0.6385 ± 0.001   | 0.2702 ± 0.0011 |
> | BetaShap(1,4)  | 2.679 ± 0.0061  | 0.6384 ± 0.0009  | 0.2706 ± 0.001  |
> | BetaShap(1,16) | 2.5865 ± 0.01   | 0.6248 ± 0.0015  | 0.2859 ± 0.0017 |
> | Random         | 2.689 ± 0.0112  | 0.6374 ± 0.0017  | 0.2712 ± 0.0019 |
> | PBanzhaf(MRLN) | 2.7396 ± 0.0022 | 0.6447 ± 0.0003  | 0.263 ± 0.0004  |
>
> Diabetes:
>
> | Name           | Logit AOPC      | Probability AOPC | IROF            |
> |----------------|-----------------|------------------|-----------------|
> | WBanzhaf(0.25) | 3.4762 ± 0.0082 | 0.7881 ± 0.001   | 0.1839 ± 0.0011 |
> | KBanzhaf       | 3.5125 ± 0.0063 | 0.7972 ± 0.0005  | 0.1745 ± 0.0005 |
> | WBanzhaf(0.75) | 3.4157 ± 0.0168 | 0.7964 ± 0.0012  | 0.1752 ± 0.0012 |
> | KSHAP          | 3.4892 ± 0.0095 | 0.7919 ± 0.0012  | 0.1798 ± 0.0012 |
> | BetaShap(16,1) | 3.3496 ± 0.0236 | 0.7596 ± 0.0058  | 0.2134 ± 0.006  |
> | BetaShap(4,1)  | 3.4245 ± 0.0168 | 0.7752 ± 0.0037  | 0.1972 ± 0.0039 |
> | BetaShap(1,4)  | 3.4372 ± 0.0169 | 0.7964 ± 0.0014  | 0.1753 ± 0.0015 |
> | BetaShap(1,16) | 3.2732 ± 0.0243 | 0.7872 ± 0.0022  | 0.1849 ± 0.0023 |
> | Random         | 3.4324 ± 0.025  | 0.7876 ± 0.003   | 0.1843 ± 0.003  |
> | PBanzhaf(MRLN) | 3.5303 ± 0.0073 | 0.7971 ± 0.0006  | 0.1746 ± 0.0007 |
>
> California Housing:
>
> | Name           | Logit AOPC      | Probability AOPC | IROF            |
> |----------------|-----------------|------------------|-----------------|
> | WBanzhaf(0.25) | 4.4203 ± 0.0068 | 0.7983 ± 0.0004  | 0.131 ± 0.0005  |
> | KBanzhaf       | 4.4382 ± 0.0049 | 0.801 ± 0.0002   | 0.128 ± 0.0002  |
> | WBanzhaf(0.75) | 4.3462 ± 0.0154 | 0.799 ± 0.0006   | 0.1301 ± 0.0006 |
> | KSHAP          | 4.4308 ± 0.0069 | 0.8002 ± 0.0004  | 0.1288 ± 0.0004 |
> | BetaShap(16,1) | 4.3148 ± 0.0215 | 0.7825 ± 0.0038  | 0.1476 ± 0.0041 |
> | BetaShap(4,1)  | 4.389 ± 0.0127  | 0.7922 ± 0.0021  | 0.1373 ± 0.0022 |
> | BetaShap(1,4)  | 4.3873 ± 0.0117 | 0.7997 ± 0.0007  | 0.1294 ± 0.0007 |
> | BetaShap(1,16) | 4.264 ± 0.021   | 0.795 ± 0.0012   | 0.1346 ± 0.0013 |
> | Random         | 4.3765 ± 0.019  | 0.7958 ± 0.0017  | 0.1334 ± 0.0018 |
> | PBanzhaf(MRLN) | 4.4485 ± 0.0059 | 0.8011 ± 0.0003  | 0.1279 ± 0.0003 |
>
>
> **Q2: Could you discuss limitations and potential pitfalls of using MRLN or similar heuristics for w_i estimation? Maybe you can include ablation studies on the impact of different w_i settings (e.g., random vs. learned) that would help validate the robustness of the method.**
>
> A: Using heuristics like MRLN has similar issues as any other heuristic methods, such as theoretical suboptimality at individual level. However, MRLN is not the core of this paper and is simply one way of computing $w_i$. If anything, the heuristic nature of MRLN further underscores the importance of feature-wise probability: even suboptimal adjustments can make statistically significant gains. As mentioned in the limitations, it would be worthwhile to find theoretically robust methods for computing $w_i$.
>
> All forms of Banzhaf are effectively different $w_i$ settings, except Weighted Banzhaf holds it at constant probability and KernelBanzhaf holds it at 0.5. I have performed tests with randomized feature weights in the comment above. Randomized weights still underperform relative to regular Banzhaf and Preference Banzhaf with MRLN.

---

> > ### Comment · Reviewer_iu3B · 2025-08-05
> >
> > Thank you for clarifying my questions. As you acknowledge that MRLN is a heuristic approach and that finding a theoretically robust method for estimating $w_i$ is a limitation and an avenue for future work, this confirms my initial concern about the difficulty of obtaining valid $w_i$ values in a non-arbitrary way for real-world applications, which is a challenge for the method's generalizability. For these reasons, I will keep my original ratings.

---

### Official Review · Reviewer_W882 · 2025-07-03

**Clarity:** 3
**Significance:** 2
**Originality:** 2
**Rating:** 3
**Confidence:** 4

**Summary:**

In explainable AI, Shapley Value and Banzhaf value arising from game theory are frequently used to quantify the importance of a feature (comparable to a player in game theory). The importance of a feature f is measured by taking all subsets (called coalitions) of other features and measuring the average marginal improvement in performance of models by adding f to those subsets.

This paper is based on the premise that not all subsets are equally likely, as ".. in real games, each player can have different preference for joining a coalition."

With this idea, the paper introduces Preference Banzhaf value, where each feature $i$ is associated with a probability weight $w_i$ of its affinity of joining a coalition. The definition (eq 4) is the natural definition with such a consideration.

The theoretical results showing the value as a solution to a regression problem and convergence closely follow those for KernelBanzhaf. I have not checked the proofs in detail, but do not see a reason to doubt them.

Experimental results compare with existing Banzhaf and Shapley value based methods. The Weights $w_i$ are set using MRLN [34].

**Questions:**

1. Can you explain what $w_i$ represents? Why is the affinity for forming a coalition a factor, and how can we know it? An example can help.
2. What is this reference [34] that you are using?
3. What is your measure of $w_i$ in experiments?
4. What do you mean by Faithfulness?
5. What is the qualitative evaluation section and Fig 4 showing?

**Ethical Concerns:**

["NO or VERY MINOR ethics concerns only"]

**Final Justification:**

Thanks to the authors for the discussion.

I feel there are some interesting points here, but the work still lacks a clear logical structure of exactly what problem and solutions are. Also, the use of the method "MRLN" makes the rigor of the paper suspect.

So I am retaining my score.

**Limitations:**

Yes.

**Paper Formatting Concerns:**

Some text elements seem to be written in math mode e.g. $KernelSHAP$, which makes them awkward to read.

**Quality:**

2

**Strengths And Weaknesses:**

Strengths:

The paper has good readability. It provides theoretical as well as experimental results. It is on a timely topic.

Weaknesses:

The main weakness is that the point of the paper is unclear. What does it means for different features to have different affinity to join coalitions and why it should this improve explanations are not discussed.

Unlike human players, features do not have personal preferences. Their utility is determined by the training algorithm. And the purpose of Banzhaf value is to elicit this utility. Manipulating the coalitions themselves (by assigning weights to features befroehand) seems circular and interfering with the objective.

In experiments, the weights $w_i$ seems to be computed using some technique in reference [34]. but I cannot find this reference, and it is not explained what this method does.

Reference [34] simply says:
Anonymous. MRLN: Adjusting masking probabilities based on model response, 2025.

The experiment section is generally poorly written. The quantities being measured and their importance are not explained. Overall, I am unconvinced that the experiments are showing something useful.

---

> ### Author Rebuttal · Authors · 2025-07-30
>
> Thank you very much for your comments. Our responses are as follows:
>
> **C1: The main weakness is that the point of the paper is unclear. What does it means for different features to have different affinity to join coalitions and why it should this improve explanations are not discussed.**
>
> **Q1: Can you explain what $w_i$ represents? Why is the affinity for forming a coalition a factor, and how can we know it? An example can help.**
>
> A: Preferences correspond to masking probabilities in terms of implementation. Intuitively, we are forcing certain masks to be selected more frequently than others to align with user objectives.
>
> For example, we can sample specific coalition structures more frequently by matching the probabilities between features in the structures. Consider a model with 6 features, and we wish to select coalition structure [1,2,3] and [4,5,6] more frequently. If we set $w_{[1,2,3]}$ to 0.2 and [3,4,5] to 0.8, the sampled coalitions are more likely to select the two coalitions separately ($0.8^6$) than together ($0.2^3*0.8^3$).
>
> Another example is reflecting domain knowledge in XAI process. If a user considers or knows feature A to be important by domain knowledge, they may mask it more frequently so that the ranking of other features (after excluding the important feature) can be captured more correctly.
>
> The original motivation can be interpreted as coalition selection that maximizes total expected payoff (sum of Banzhaf values).
>
> **C2: Unlike human players, features do not have personal preferences. Their utility is determined by the training algorithm. And the purpose of Banzhaf value is to elicit this utility. Manipulating the coalitions themselves (by assigning weights to features befroehand) seems circular and interfering with the objective.**
>
> A: The idea of 'personal preference' is an analogy for our motivation - that each feature should use different masking probability. As discussed in the previous comment, the probability itself is determined by the user's objectives.
>
> **C3: In experiments, the weights seems to be computed using some technique in reference [34]. but I cannot find this reference, and it is not explained what this method does. Reference [34] simply says: Anonymous. MRLN: Adjusting masking probabilities based on model response, 2025.**
>
> **Q2: What is this reference [34] that you are using?**
>
> A: The main idea of MRLN is to (a) compute the model behavior (either as output or latent vector) from a uniform perturbation; (b) select outputs satisfying locality criteria in the behavior space; (c) compute empirical probability based on the masks of the selected outputs; and (d) perform linear regression with the new probability. This method heuristically aligns mask probability with model’s empirical behavior, making it suitable as an example for selecting masking probabilities when a user’s objective is to improve explanation fidelity.
>
> **C4: The experiment section is generally poorly written. The quantities being measured and their importance are not explained. Overall, I am unconvinced that the experiments are showing something useful.**
>
> **Q4: What do you mean by Faithfulness?**
>
> A: The quantities are traditionally used metrics for measuring fidelity and sensitivity, so their descriptions had been removed due to text limit.
>
> In XAI, we typically want the explanations to satisfy several desiderata. Two typical properties discussed in the literature are faithfulness and sensitivity. Faithfulness (also known as fidelity) is the degree to which an explanation matches the model output behavior. Ideally, more important features would have greater impact on model decisions. Sensitivity (also known as robustness or consistency) measures the stability of the attribution for identical inputs. A good explanation should be stable, i.e., have lower sensitivity.
>
> In this paper, we considered Area over Perturbation Curve (AOPC) and Iterative Removal of Features (IROF) for faithfulness. AOPC measures the area over the curve (AOC) of the change in model output from the original as we replace each feature in the input with the chosen baseline:
>
> $$
> AOPC = \frac{1}{L+1}\sum^{L}_{k=1}{(f(x^0_M)-f(x^k_M))}
> $$
>
> Where $M$ is the feature replacement procedure and $L$ is the number of replacements. For classifiers, $f$ is usually the logit or probability of the target class. IROF follows a similar process, except that it works exclusively with probabilities:
>
> $$
> IROF = \frac{1}{L+1}\sum^{L}_{k=1}{(p(x^k_M)/p(x^0_M))}
> $$
>
> A model is more faithful if it has higher AOPC and lower IROF.
>
> For sensitivity, we evaluate $L_2$-normalized error between attributions across several runs and the true value. If the true value is difficult to calculate, we use the average as the proxy.
>
> $$
> err_{l2} = \frac{1}{|\bar{\beta}|^2_2}\sum^{L}_{k=1}|\beta_k-\bar{\beta}|_2^2
> $$
>
> We also use average pairwise rank correlation and top-K Jaccard distance amongst attributions.
>
> $$
> \rho = \frac{1}{L-1}\sum^{L-1}_{k=1}{corr( \beta_k, \beta_k + 1 )}
> $$
>
> and
>
> $$
> J_K = \frac{1}{L-1}\sum^{L-1}_{k=1}{1-\frac{|topk(\beta_k)\cap topk(\beta_k+1)|}{|topk(\beta_k)\cup topk(\beta_k+1)|}}
> $$
>
>
> where $L$ is the number of times the explanation has been evaluated. A method is less sensitive if it has lower $L_2$-normalized error, higher correlation, and higher Jaccard index.
>
>
>
> **Q3: What is your measure of in experiments?**
>
> A: $w_i$ is measured in decimals since it is the masking probabilities.
>
> **Q5: What is the qualitative evaluation section and Fig 4 showing?**
>
> A: The qualitative section shows that aligning $w_i$ to 'important sections' of the input creates explanations with greater fidelity. For images, the important section is the object. Figure 4 is meant to show that the attribution is more faithful (i.e., highlights the real image better) when $w_i$ is reduced for sections that overlap with the object. This aligns with the notion that we can find more accurate rankings for less-important segments by removing the more-important segments during their calculations. It is analogous to reflecting domain knowledge in the probabilities.

---

> ### Comment · Reviewer_W882 · 2025-08-04
>
> I read the authors comments but I still do not get the point of the paper and what it is trying to do.
>
> The authors can answer if they wish,  but
> I am not sure that continuing this discussion is useful.

---

> > ### Author Response · Authors · 2025-08-04
> >
> > Thank you very much for your reply.
> >
> > If you could let me know which part is still confusing, it would be greatly appreciated since it would help us further improve the paper.
> >
> > For the original paper and our response above, they can be summarized as as follows:
> > - We can change the masking probabilities by feature when using Banzhaf values, and should change them to better align the explanations to the user's requirements
> > - A 'good set of probabilities' would depend on the user's needs. For example, using regular Banzhaf value ($w_i=0.5 \forall i$) would be best if the user wants the highest theoretical stability. However, it would be better to adjust the probabilities if the objective is higher fidelity.
> > - Techniques for computing optimal $w_i$ for greater fidelity is not the main objective of this paper; however, we show that even heuristic methods can improve explanation faithfulness.
> > - We show that when faithfulness is the user's objective, $w_i$ represents 'a user's a priori belief on a feature's importance'. For images, it would be the location of the main object. This means a user may reflect a priori information like domain knowledge by adjusting $w_i$ accordingly.

---

### Official Review · Reviewer_oua1 · 2025-07-06

**Clarity:** 1
**Significance:** 1
**Originality:** 1
**Rating:** 3
**Confidence:** 4

**Summary:**

The paper introduces the so-called Preference Banzhaf, a game-theoretic/perturbation based XAI alternative (following the SHAP value line), in particular a modification of the solution concept Banzhaf (in cooperative game theory, dropping the efficiency axiom) w.r.t. preference/likelihood of the feature taking place in the coalition of features. Authors show that their approach shows  favourable performance compared to KernelSHAP (a common variation of SHAP) and other Banzhaf-based methods. These empirical results (synthetic + real world datasets) confirm the theoretical results that authors introduce, basically convergence to centred linear regression (w/ and wo/ the intercept), and the true values. Here, implicitly the work is also inspired by LIME in terms of linear regression.

**Questions:**

Q1: What would these preferences correspond in XAI setting? (since features themselves cannot "want" to be with some other features). Are redundant or proxy features (always some taking place together) occurring with each other be considered as an example? But then how  can we justify the so-called "preferences"? (As in some case, I would argue that different coalitions would make a better coverage for the feature importance, so stop say, should we consider rather low probability ones?).

Q2: Related to earlier question: how do we get these preferences in XAI setting right? You are using MLRN, but what does convergence to True value  mean?

Q3: How do you respond to my comment 5 (C5). Also, I don't know what to do with Figure 4, do they look great?  (columns labels etc also missing, only explained in the caption, what is the legend? etc.  )

**Ethical Concerns:**

["NO or VERY MINOR ethics concerns only"]

**Final Justification:**

I appreciate the effort of answers from the author(s). Now, I understand the paper better e.g., how the preferences play a role in terms of reflecting the background knowledge of the user. ( which I see other reviewers were also confused about the intentions of the paper.) I will increase my score half a point. This also reflects my current position: While still having the doubts about the depth and substance of the contribution (theoretical results for instance), I am certain that the experimental section quality in its current version, not "acceptable" to be published at the conference. Overall, my warm suggestion for authors is to improve the exposition of their work greatly, and try again so that their contribution becomes more clear.

**Limitations:**

Yes.

**Paper Formatting Concerns:**

I don't have any.

**Quality:**

1

**Strengths And Weaknesses:**

Strengths:

- The direction of research is relevant, in particular, considering the influence of game-theory inspired XAI techniques, both in academia and the industry. So it would be significant direction.

Weakness:

There are plenty weaknesses:

W1: In terms of clarity, lots of details are not nicely explained, especially formalisms and crucial technical values. See detailed comments below.
W2: I wonder how theorem 1 and theorem 2 are not redundant, since one is with intercept (why do you need another theorem for this. )

W3: regarding originality and quality, contribution seems to be incremental for this type of venue.



More detailed comments are as follows by my comments enumerated through Cs:

C1: the motivation of their direction 46-54 is framed in terms of game theory solely, but it should be based on XAI, since you are not contributing to game theory but XAI. See Q1 below.

C2:  in line 44 you don't need to say excluding said player, since this is the same for even classical Shapley value. (since it measures the player contribution). But you need to explain the Banzhaf value better.

C3: In related work, you have a section on LIME, but how is this relevant  for your work (my guess, centred linear regression?) but this is not motivated. Also KernelBanzhaf seems to be highly relevant related work for your work, but you don't cite it in related work section, but only in methods.

C4: Lots of technical details are not explained greatly.  w values,  GLIME,  what does these v^2 \gamma^4 mean, and almost all the other constants/variables. T/here is basically no intuition provided. (e.g., M is just some constant, is it negative? is it positive? why it exist? what is a good value? is it important? how influential it is etc.) Similar for d delta choice of epsilon w_i, the powers of these terms etc.

C5: I am having hard time to justify why is this an important contribution, and you don't seem to explain it much either. Given the results, there is not much added value, then conceptually what should be the justification is not clear to me.  (Side comment: In the tables, you did not show Preference Banzhaf, but MLRN, which is also a bit confusing) These tables are not very informative.  Also, I don't know what to do with Figure 4, do they look great? (also mentioned in Question 3). In its current form, these look pretty weak in being informative (as the major rest of the paper).

Minor issues:



m1: typo in 39: "KernelSHAP is suffers..."

m2: "One issue with Shapley" (I guess you mean Shapley value for KernelSHAP, must be indicated)

m3: typing up the paranthesis of Equation 4 (preferably not squared ones but bigger normal parentheses)


Overall, my warm recommendation to authors would be, to submit their work to dedicated decent venues (such XAI world conference), where the value of their contribution could be better appreciated, and hopefully get also further constructive feedback.

---

> ### Author Rebuttal · Authors · 2025-07-30
>
> Thank you very much for your review and recommendations. Our responses are as follows:
>
> **W2: I wonder how theorem 1 and theorem 2 are not redundant, since one is with intercept (why do you need another theorem for this.)**
>
> A: The presence of the intercept changes the closed-form formula for the feature coefficients. What I have shown is that, specifically for the setup where the inputs are binary, centering with or without intercept results in identical expected solutions. This nuance is important because implementation-wise, excluding the intercept makes the linear surrogate sensitive to vertical shifts (such as subtracting a baseline value). Consequently, we not only need to explicitly set f(0)=0 for Theorem 1, and it is no longer equivalent to uncentered linear regression.
>
> **W3: regarding originality and quality, contribution seems to be incremental for this type of venue.**
>
> A: If you could elaborate on this comment, it would be greatly appreciated. As far as I am aware, this approach is the first to show that we can modify probabilities for individual features and still achieve axiomatic properties, unlike existing methods that apply the same probability across all features. It is also the first to show empirically that it is desirable to make feature-wise adjustment, as we can achieve statistically significantly greater faithfulness even with heuristically chosen probabilities (MRLN).
>
> **C1: the motivation of their direction 46-54 is framed in terms of game theory solely, but it should be based on XAI, since you are not contributing to game theory but XAI. See Q1 below.**
>
> **Q1: What would these preferences correspond in XAI setting? (since features themselves cannot "want" to be with some other features). Are redundant or proxy features (always some taking place together) occurring with each other be considered as an example? But then how can we justify the so-called "preferences"? (As in some case, I would argue that different coalitions would make a better coverage for the feature importance, so stop say, should we consider rather low probability ones?).**
>
> A: Preferences correspond to masking probabilities in terms of implementation. Intuitively, we are forcing certain masks to be selected more frequently than others. While features themselves cannot ‘want’ to be with other features, there would be ‘better’ coalitions that a user would like to select from in order to achieve specific objectives.
>
> For example, we can sample specific coalition structures more frequently by matching the probabilities between features in the structures. Consider a model with 6 features, and we wish to select coalition structure [1,2,3] and [4,5,6] more frequently. If we set $w_{[1,2,3]}$ to 0.2 and [3,4,5] to 0.8, the sampled coalitions are more likely to select the two coalitions separately ($0.8^6$) than together ($0.2^3*0.8^3$).
>
> Another example is reflecting domain knowledge in XAI process. If a user considers or knows feature A to be important by domain knowledge, they may mask it more frequently so that the ranking of other features (after excluding the important feature) can be captured more correctly.
>
> The original motivation can be interpreted as coalition selection that maximizes total expected payoff (sum of Banzhaf values).
>
> **C2: in line 44 you don't need to say excluding said player, since this is the same for even classical Shapley value. (since it measures the player contribution). But you need to explain the Banzhaf value better.**
>
> A: We clarified the explanation for Banzhaf values.
>
> **C3: In related work, you have a section on LIME, but how is this relevant for your work (my guess, centred linear regression?) but this is not motivated. Also KernelBanzhaf seems to be highly relevant related work for your work, but you don't cite it in related work section, but only in methods.**
>
> A: We included the section on LIME as it can be considered the basis for kernelized game-theoretic approaches. The latter was indeed an error on our part as it was meant to be included when discussing Banzhaf values.
>
> **C4: Lots of technical details are not explained greatly. w values, GLIME, what does these v^2 \gamma^4 mean, and almost all the other constants/variables. T/here is basically no intuition provided. (e.g., M is just some constant, is it negative? is it positive? why it exist? what is a good value? is it important? how influential it is etc.) Similar for d delta choice of epsilon w_i, the powers of these terms etc.**
>
> A:
>
> 1.The definitions for $v^2$ and $\gamma^2$ are given in the convergence theorem as $max(w_i(1-w_i))$ and $\sum{(1/w_i(1-w_i))}$. Intuitively, $v^2$ is the maximum amongst the variance of the masks, while $\gamma^2$ is the sum of their precisions (inverse of variance).
>
> 2. We showed $v^2 \gamma^4$ as a part of the results because the two variables are connected by w and often move concurrently.
>
> 3. M is the bound on f(z).
>
> 4. delta and epsilon are positive constant representing probability and L2 error tolerances, which are often used in convergence.
>
> **C5: I am having hard time to justify why is this an important contribution, and you don't seem to explain it much either. Given the results, there is not much added value, then conceptually what should be the justification is not clear to me. (Side comment: In the tables, you did not show Preference Banzhaf, but MLRN, which is also a bit confusing) These tables are not very informative. Also, I don't know what to do with Figure 4, do they look great? (also mentioned in Question 3). In its current form, these look pretty weak in being informative (as the major rest of the paper).**
>
> **Q3: How do you respond to my comment 5 (C5). Also, I don't know what to do with Figure 4, do they look great? (columns labels etc also missing, only explained in the caption, what is the legend? etc. )**
>
> A:
>
> 1. We used MRLN as it can automatically determine w_i based on model-based locality. We reported MRLN because it was the algorithm being used, but I understand that it may have been better to clarify it like PBanzhaf(MRLN).
>
> 2. These tables mainly show that choosing feature-wise w_i can help achieve a user’s objective. For example, MRLN-based probability selection makes statistically significant gains in fidelity based on difference-of-mean tests. This is despite MRLN being heuristic and is likely suboptimal, suggesting that even minor adjustments can make significant improvements. As the tables are difficult to interpret at first glance, it may have been better to present them as graphs where the differences and their significance are far more visible.
>
> 3. Figure 4 was meant to show that the importance ranks of segments change significantly based on object’s location when we give low probability on the center or the periphery. In a sense, we are inserting ‘domain knowledge’ regarding the location of the object by adjusting the probabilities.
>
> **Q2: Related to earlier question: how do we get these preferences in XAI setting right? You are using MLRN, but what does convergence to True value mean?**
>
> A:
>
> 1. As discussed above, the preferences are determined by the user's needs. If a user needs better fidelity, they may use methods like MRLN to modify the probabilities to remove more relevant features more frequently. They may choose to change probabilities to manually reflect domain knowledge. If a user needs higher stability, they would use traditional Banzhaf as it has the best theoretical convergence.
>
> 2. In the case of MRLN, the True value would be the expected contribution of individual features based on coalitions within the locality defined by the model’s behavior. However, MRLN is simply a method for choosing $w_i$, not the main focus of this paper.

---

> > ### Comment · Reviewer_oua1 · 2025-08-05
> > **After rebuttal**
> >
> > I appreciate the effort of answers from the author(s). Now, I understand the paper better e.g., how the preferences play a role in terms of reflecting the background knowledge of the user. ( which I see other reviewers were also confused about the intentions of the paper.) I will increase my score half point. This also reflects my current position: While still having the doubts the depth and substance of the contribution (theoretical results for instance), I am certain that the experimental section quality in its current version, not "acceptable" to the conference. Overall, my warm suggestion for authors is to improve the exposition of their work greatly, and try again so that their contribution becomes more clear.

---

### Decision · Program_Chairs · 2025-09-17

**Decision:**

Reject

**Comment:**

## Summary
The paper  draws on literature arguing that the Banzhaf value is a more stable variant of the Shapley value (obtained by dropping an axiom), but that it is limited to treat all "features" the same when calculating the influence of a feature. The paper proposes a weighted version of the Banzhaf value and shows how it can be computed (given the weights).
## Strengths
In general the reviewers appreciated the topic of study and the explorations of game-theory inspired work in interpretability, given the popularity of methods like SHAP.

## Weaknesses
Reviewers identified a number of weaknesses.
1. The main theoretical results are already known (the reviewer provided cites).
2. It's unclear how to produce the "weights" that the weighted Banzhaf requires, and what the justification of these weights might look like.
3. There was some concern about the experiments as presented, and what story they are trying to tell.

## Author Discussion

There was a robust discussion between authors and reviewers on these issues. In particular
1. The authors acknowledged the references and promised to include them. They argued that some value still remains in their analysis, but I'm skeptical.
2. They provided an argument for how to interpret weights, but their examples were confusing (at least to me) since they defined weights of coalitions, but in the paper weights are defined in terms of individual features as "what's the probability this feature joins ANY coalition". They rely on a different paper that is listed as Anonymous in the references as a way to generate the weights, and all of this seems very confusing and murky.
3. Concerns about the presentation of experiments were addressed: the authors even ran some new experiments in response to reviewer requests, which I appreciate.

## Justification
In my view, there's a lot of work to be done to argue for the value of the preference-banzhaf. It's a good initial direction, but it will be important for the authors to make a stronger case for where the weights might come from or how one might intuitively think about weights for features.